# Towards Painless Policy Optimization for Constrained MDPs

**Arushi Jain**[1][*]    **Sharan Vaswani**[2][*]    **Reza Babanezhad**[3]    **Csaba Szepesvári**[4,5]    **Doina Precup**[1,5]

[1]Mila, McGill University
[2]Simon Fraser University
[3]SAIT AI Lab, Montreal
[4]Amii, University of Alberta
[5]DeepMind

## Abstract

We study policy optimization in an infinite horizon, $\gamma$-discounted constrained Markov decision process (CMDP). Our objective is to return a policy that achieves large expected reward with a small constraint violation. We consider the online setting with linear function approximation and assume global access to the corresponding features. We propose a generic primal-dual framework that allows us to bound the reward sub-optimality and constraint violation for arbitrary algorithms in terms of their primal and dual regret on online linear optimization problems. We instantiate this framework to use coin-betting algorithms and propose the **Coin Betting Politex (CBP)** algorithm. Assuming that the action-value functions are $\varepsilon_{\mathrm{b}}$-close to the span of the $d$-dimensional state-action features and no sampling errors, we prove that $T$ iterations of CBP result in an $O\left(\frac{1}{(1-\gamma)^3\sqrt{T}} + \frac{\varepsilon_{\mathrm{b}}\sqrt{d}}{(1-\gamma)^2}\right)$ reward sub-optimality and an $O\left(\frac{1}{(1-\gamma)^2\sqrt{T}} + \frac{\varepsilon_{\mathrm{b}}\sqrt{d}}{1-\gamma}\right)$ constraint violation. Importantly, unlike gradient descent-ascent and other recent methods, CBP does not require extensive hyperparameter tuning. Via experiments on synthetic and Cartpole environments, we demonstrate the effectiveness and robustness of CBP.

## 1 INTRODUCTION

Popular reinforcement learning (RL) algorithms focus on optimizing an unconstrained objective, and have found applications in games such as Atari (Mnih et al., 2015) or Go (Silver et al., 2016), robot manipulation tasks (Tan et al., 2018; Zeng et al., 2020) or clinical trials (Schaefer et al., 2005). However, many applications require the planning agent to satisfy constraints – for example, in wireless sensor networks (Buratti et al., 2009; Julian et al., 2002) there is a constraint on average power consumption of a deployed policy. Similarly, in safe RL, the policy is constrained to only visit certain states while exploring in physical systems (Moldovan and Abbeel, 2012; Ono et al., 2015; Fisac et al., 2018). The constrained Markov decision process (CMDP) (Altman, 1999) is a natural framework to model long-term constraints that need to be satisfied by a policy. The typical objective for CMDPs is to maximize the cumulative reward (similar to unconstrained MDPs), while (approximately) satisfying the constraint.

We focus on a well-studied problem in CMDPs – return an approximately feasible policy (that is allowed to violate the constraints by a small amount), while (approximately) maximizing the cumulative reward. The past literature on this topic considered two approaches. The first approach is *primal-only algorithms*, where constraints are (approximately) enforced without directly relying on introducing a Lagrangian formulation (Achiam et al., 2017; Chow et al., 2018; Dalal et al., 2018; Liu et al., 2020; Xu et al., 2021). Of these methods, only the recent work of Xu et al. (2021) guarantees global convergence to the optimal feasible policy in both the tabular and function approximation settings.

The second approach in CMDPs is to form the Lagrangian, and solve the resulting saddle-point problem using *primal-dual algorithms* (Altman, 1999; Borkar, 2005; Bhatnagar and Lakshmanan, 2012; Borkar and Jain, 2014; Tessler et al., 2018; Liang et al., 2018; Paternain et al., 2019; Yu et al., 2019; Ding et al., 2021, 2020; Stooke et al., 2020). Such approaches update both the policy parameters (primal variables), while updating the Lagrange multipliers (dual variables). Of

---

[*]The first two authors contributed equally. Email: arushi.jain@mail.mcgill.ca, vaswani.sharan@gmail.com.

*Accepted for the 38th Conference on Uncertainty in Artificial Intelligence* (UAI 2022).

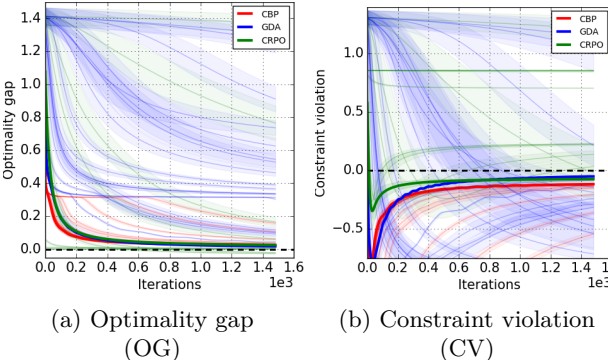

(a) Optimality gap
(OG)

(b) Constraint violation
(CV)

Figure 1: **Hyperparameter sensitivity:** Optimality gap and constraint violation (averaged across 5 runs) for different hyperparameters for **GDA** (Ding et al., 2020), **CRPO** (Xu et al., 2021) and the proposed algorithm **CBP** on a gridworld environment with access to the true CMDP. The dark lines show the performance of the best hyperparameter, while the lighter-shade lines represent results while using other hyperparameters. Both GDA and CRPO exhibit large variations in their performance, while CBP is more robust. See Section 6 for details.

these methods, Tessler et al. (2018) prove a local convergence guarantee, while Paternain et al. (2019) prove that their proposed algorithm will converge to a neighbourhood of the optimal policy. More recently, Ding et al. (2020) proposed to use natural policy gradient updates (Kakade, 2001) for changing the policy parameters while using gradient descent to update the dual variables. They prove that this primal-dual algorithm converges to the optimal policy in both the tabular and the function approximation settings.

Although there is no lack in algorithms designed for CMDPs, *these algorithms are often highly sensitive to the choice of their hyperparameters.* For example, Figure 1 demonstrates the effect of varying the hyperparameters for two provably efficient algorithms, the primal-dual natural-policy ascent, gradient descent method (in short, GDA) of Ding et al. (2020) and the primal-only CRPO method of Xu et al. (2021) on a synthetic tabular environment. While one can find hyperparameters that control the worst-case performance of either GDA or CRPO, such choices result in a poor empirical performance on individual instances, a feature that GDA and CRPO share with unconstrained MDP policy optimization algorithms, such as Politex (Abbasi-Yadkori et al., 2019), or natural policy gradient (Kakade, 2001).

**CONTRIBUTIONS:** *Designing robust policy optimization algorithms that require minimal hyperparameter tuning is our main motivation, and towards this, we make the following contributions.*

**Generic Primal-Dual Framework**: In Section 3, we cast the problem of planning in discounted infinite horizon CMDPs to a generic primal-dual framework. In particular, we prove that any algorithm that can control (i) the primal and dual regret for specific online linear optimization problems and (ii) the errors due to function approximation and sampling, will (approximately) maximize the cumulative discounted reward while (approximately) minimizing the constraint violation (Theorem 3.1). Importantly, this result holds for any CMDP and is independent of how the policies or value functions are represented.

**Instantiating the Framework**: In Section 4, we instantiate the framework using two algorithms from the online linear optimization literature – Gradient Descent Ascent (GDA) (Section 4.2.1) and Coin-Betting (CB) (Section 4.2.2). While GDA requires setting the hyperparameters to specific problem-dependent constants, CB is more robust to hyperparameter tuning (see Figure 1). In the simpler tabular setting, the approximation errors can be easily controlled and we use Theorem 3.1 in conjunction with existing regret bounds to prove that the average optimality gap (difference in the cumulative reward of achieved policy and the optimal policy) and the average constraint violation decrease at an $O\left(1/\sqrt{T}\right)$ rate (Corollaries A.1 and A.2).

**Handling Linear Function Approximation**: In Section 5, we assume global access to a $d$-dimensional feature map $\Phi : \mathcal{S} \times \mathcal{A} \to \mathbb{R}^d$, and that the action-value functions for any policy are $\varepsilon_{\mathrm{b}}$-close to the span of these features. With this assumption, we prove that it is possible to control the approximation errors for each state-action pair. Subsequently, in Section 5.1, we use the robust coin-betting algorithms to instantiate the primal-dual framework in the linear function approximation setting and propose the *Coin-Betting Politex* (CBP) algorithm. Ignoring sampling errors, in Section 5.1.1, we prove that the average optimality gap for CBP scales as $O\left(\frac{1}{(1-\gamma)^3 \sqrt{T}} + \frac{\varepsilon_{\mathrm{b}}\sqrt{d}}{(1-\gamma)^2}\right)$, while the average constraint violation is $O\left(\frac{1}{(1-\gamma)^2 \sqrt{T}} + \frac{\varepsilon_{\mathrm{b}}\sqrt{d}}{(1-\gamma)}\right)$. With linear function approximation, the average constraint violation for the algorithm of Ding et al. (2020) decreases at a worse $O\left(1/T^{1/4}\right)$ rate. On the other hand, the CRPO algorithm of Xu et al. (2021) results in an $O\left(1/\sqrt{T}\right)$ bound for both the average suboptimality and constraint violation. However, both algorithms can amplify the function approximation errors to large, potentially unbounded values. Importantly, both algorithms require typically unknown quantities which impedes their practical use.

**Experimental Evaluation**: In Section 6, we first describe some practical considerations when implementing CBP. We then evaluate CBP and compare its em-

pirical performance to the algorithms of Ding et al. (2020); Xu et al. (2021). Our experiments on synthetic tabular environment and the Cartpole environment with linear function approximation demonstrate the consistent effectiveness and robustness of CBP.

## 2 PROBLEM FORMULATION

We consider an infinite-horizon discounted constrained Markov decision process (CMDP) (Altman, 1999) defined by the tuple $\langle \mathcal{S}, \mathcal{A}, \mathcal{P}, r, c, b, \rho, \gamma \rangle$ where $\mathcal{S}$ is the countable set of states, $\mathcal{A}$ is the countable action set, $\mathcal{P} : \mathcal{S} \times \mathcal{A} \to \Delta_{\mathcal{S}}$ is the transition probability function, $\Delta_{\mathcal{S}}$ is the $\mathcal{S}$-dimensional probability simplex, $\rho \in \Delta_{\mathcal{S}}$ is the initial distribution of states and $\gamma \in [0, 1)$ is the discount factor. The primary reward to be maximized is denoted by $r : \mathcal{S} \times \mathcal{A} \to [0, 1]$. For each state $s$, we define the reward value function w.r.t. the policy $\pi : \mathcal{S} \to \Delta_{\mathcal{A}}$ as $V_r^\pi(s) = \mathbb{E}_{\pi, \mathcal{P}} \left[ \sum_{t=0}^\infty \gamma^t r(s_t, a_t) | s_0 = s \right]$ where $a_t \sim \pi(\cdot | s_t)$, and $s_{t+1} \sim \mathcal{P}(\cdot | s_t, a_t)$ and $\Delta_{\mathcal{A}}$ is $\mathcal{A}$-dimensional simplex. The expected discounted return or *reward value* of a policy $\pi$ is defined as $V_r^\pi(\rho) = \mathbb{E}_{s_0 \sim \rho} \left[ V_r^\pi(s_0) \right]$. Similarly, the constraint reward is denoted by $c : \mathcal{S} \times \mathcal{A} \to [0, 1]$ and the *constraint reward value* for $\pi$ by $V_c^\pi(\rho)$. For each $(s, a)$ under policy $\pi$, the reward action-value function is defined as $Q_r^\pi : \mathcal{S} \times \mathcal{A} \to \mathbb{R}$ s.t. $Q_r^\pi(s, a) = r(s, a) + \gamma \mathbb{E}_{s' \sim \mathcal{P}(\cdot | s, a)}[V_r^\pi(s')]$ and satisfies the relation: $V_r^\pi(s) = \langle \pi(\cdot | s), Q_r^\pi(s, \cdot) \rangle = \mathbb{E}_{a \sim \pi(\cdot | s)}[Q_r^\pi(s, a)]$. We define $Q_c^\pi$ analogously. The agent's objective is to return a policy $\pi$ that maximizes $V_r^\pi(\rho)$, while ensuring that $V_c^\pi(\rho) \geq b$. Formally,

$$\max_\pi V_r^\pi(\rho) \quad \text{s.t.} \quad V_c^\pi(\rho) \geq b. \quad (1)$$

Throughout, we will assume the existence of a feasible policy (i.e., one with $V_c^\pi(\rho) \geq b$), and denote the optimal feasible policy by $\pi^*$. Due to sampling and other errors, we will aim for finding policy $\pi$ with some $\varepsilon > 0$ such that,

$$V_r^\pi(\rho) \geq V_r^{\pi^*}(\rho) - \varepsilon \quad \text{s.t.} \quad V_c^\pi(\rho) \geq b - \varepsilon. \quad (2)$$

In the next section, we specify a generic primal-dual framework solving the problem in Equation (2).

## 3 PRIMAL-DUAL FRAMEWORK

By Lagrangian duality, $\pi^*$ is a solution to Equation (1) if and only if for some $\lambda^* \geq 0$, $(\pi^*, \lambda^*)$ solves the saddle-point problem

$$\max_\pi \min_{\lambda \geq 0} V_r^\pi(\rho) + \lambda[V_c^\pi(\rho) - b]. \quad (3)$$

Here, $\lambda \in \mathbb{R}$ is the Lagrange multiplier for the constraint.

We will solve the above primal-dual saddle-point problem iteratively, by alternatively updating the policy (primal variable) and the Lagrange multiplier (dual variable). If $T$ is the total number of iterations, we define $\pi_t$ and $\lambda_t$ to be the primal and dual iterates for $t \in [T] := \{1, \ldots, T\}$. Updating the $(\pi_t, \lambda_t)$ variables will require estimating the action-value functions. We define $\hat{Q}_r^t := \hat{Q}_r^{\pi_t}$ and $\hat{Q}_c^t := \hat{Q}_c^{\pi_t}$ as the *estimated* action-value functions corresponding to the policy $\pi_t$. We also define $\hat{V}_c^\pi(s) := \langle \pi(\cdot | s), \hat{Q}_c^t(s, \cdot) \rangle$, $V_r^t(\rho) := V_r^{\pi_t}(\rho)$ and $V_c^t(\rho) := V_c^{\pi_t}(\rho)$. In this section, we assume that $\left\| Q_r^t - \hat{Q}_r^t \right\|_\infty \leq \tilde{\varepsilon}$ and $\left\| Q_c^t - \hat{Q}_c^t \right\|_\infty \leq \tilde{\varepsilon}$.

Given a generic primal-dual algorithm, our task is to characterize its performance in terms of its cumulative reward and constraint violation. Specifically, for a sequence of policies $\{\pi_0, \pi_1, \ldots, \pi_{T-1}\}$ and Lagrange multipliers $\{\lambda_0, \lambda_1, \ldots, \lambda_{T-1}\}$ generated by an algorithm, we define the *average optimality gap* (OG) and the *average constraint violation* (CV) as,

$$\text{Avg. optimality gap (OG)} := \frac{1}{T} \sum_{t=0}^{T-1} [V_r^{\pi^*}(\rho) - V_r^t(\rho)]$$

$$\text{Avg. constraint violation (CV)} := \frac{1}{T} \left[ \sum_{t=0}^{T-1} b - V_c^t(\rho) \right]_+$$

where $[x]_+ = \max\{x, 0\}$. For this algorithm, we define the *primal regret* and *dual regret* as follows:

$$\mathcal{R}^p(\pi^*, T) := \sum_{t=0}^{T-1} \Big\langle \pi^*(\cdot | s) - \pi_t(\cdot | s), \quad (4)$$

$$\hat{Q}_r^t(s, \cdot) + \lambda_t \hat{Q}_c^t(s, \cdot) \Big\rangle_{s \sim \nu_{\rho, \pi^*}},$$

$$\mathcal{R}^d(\lambda, T) := \sum_{t=0}^{T-1} (\lambda_t - \lambda) \left( \hat{V}_c^t(\rho) - b \right). \quad (5)$$

Here, $\langle f, g \rangle_{s \sim \nu_{\rho, \pi^*}} = \mathbb{E}_{s \sim \nu_{\rho, \pi^*}}[f(s)g(s)]$ and $\nu_{\rho, \pi^*}$ is the discounted occupation measure induced by following $\pi^*$ from $\rho$ normalized so that it becomes a probability measure. Observe that the above quantities correspond to the regret for online linear optimization algorithms that can independently update the primal and dual variables.[1] Our main result (proved in Appendix B) in this section characterizes the performance of a generic algorithm in terms of its primal and dual regret.

**Theorem 3.1.** *Assuming that* $\left\| Q_r^t - \hat{Q}_r^t \right\|_\infty \leq \tilde{\varepsilon}$ *and* $\left\| Q_c^t - \hat{Q}_c^t \right\|_\infty \leq \tilde{\varepsilon}$, *for a generic algorithm producing a sequence of polices* $\{\pi_0, \pi_1, \ldots, \pi_{T-1}\}$ *and dual variables* $\{\lambda_0, \lambda_1, \ldots, \lambda_{T-1}\}$ *such that for all $t$, $\lambda_t$ is constrained to lie in the $[0, U]$ where $U > \lambda^*$, OG and CV*

---

[1]Note: We are computing the primal regret with respect to the optimal policy $\pi^*$ that satisfies the constraints. $R^p(\pi^*, T)$ can be negative since the current policy $\pi_t$ can violate the constraints and obtain higher rewards.

can be bounded as:

$$\mathcal{OG} \leq \frac{\mathcal{R}^p(\pi^*, T) + (1-\gamma)\mathcal{R}^d(0, T)}{(1-\gamma)T} + \tilde{\varepsilon}\, g(U),$$

$$\mathcal{CV} \leq \frac{\mathcal{R}^p(\pi^*, T) + (1-\gamma)\mathcal{R}^d(U, T)}{(U-\lambda^*)(1-\gamma)T} + \frac{\tilde{\varepsilon}\, g(U)}{(U-\lambda^*)},$$

where $g(U) := \left[\frac{1+U}{1-\gamma} + U\right]$.

We note that such a general primal-dual regret decomposition for convex MDPs (including CMDPs) was recently done by Zahavy et al. (2021). However, they handle the tabular setting where the primal variables correspond to state-action occupancy measures, whereas, the above result defines the primal variables to be the policy parameters. More importantly, our result does not require any assumption about the underlying CMDP. In the unconstrained setting, reducing the policy optimization problem to that of online linear optimization has been previously explored in the *Politex* algorithm (Abbasi-Yadkori et al., 2019), and we build upon this work. Politex is an iterative policy optimization algorithm where the policy at each timestep is proportional to the softmax over the sum of all the action-value functions seen in the past. This algorithm bounds the optimality gap in terms of approximation error in the action-value functions and a regret term similar to online linear optimization.

In order to bound the average reward optimality gap and the average constraint violation, we need to (i) project the dual variables onto the $[0, U]$ interval and ensure that $U > \lambda^*$, (ii) update the primal and dual variables to control the respective regret in Equation (5), and (iii) control the approximation error $\tilde{\varepsilon}$. Next, we use this recipe to design algorithms with provable guarantees.

# 4  INSTANTIATING THE FRAMEWORK

In this section, we will instantiate the primal-dual framework by using the above technique – specifying the value of $U$ in Section 4.1 and describing algorithms that control the primal and dual regret in Section 4.2.

## 4.1  UPPER-BOUND FOR DUAL VARIABLES

In Appendix B, we prove the following upper-bound on the optimal dual variable

**Lemma 4.1.** *The objective Equation (1) satisfies strong duality, and the optimal dual variables are bounded as $\lambda^* \leq \frac{1}{\zeta(1-\gamma)}$, where $\zeta := \max_\pi V_c^\pi(\rho) - b > 0$.*

Unlike Ding et al. (2020, 2021) who bound the dual variables in terms of the unknown Slater constant, the upper-bound from Lemma 4.1 can be computed by maximizing the constraint value function as an unconstrained problem. Throughout, we will set $U = \frac{2}{\zeta(1-\gamma)}$ that satisfies the requirement $U > \lambda^*$ and projects the dual variables onto $[0, U]$ range (in Section 3).

## 4.2  CONTROLLING THE PRIMAL AND DUAL REGRET

In this section, we specify two algorithms to update the primal and dual variables, and control the primal and dual regret respectively. In particular, in Section 4.2.1, we will use mirror ascent to update the primal variables, and gradient descent to update the dual variables. Inspired by the literature on online linear optimization (Orabona and Pal, 2016), we will use robust, parameter-free algorithms to update the primal and dual variables in Section 4.2.2.

### 4.2.1  Gradient descent ascent

At iteration $t \in [T]$, if the primal and dual iterates are $\pi_t$ and $\lambda_t$ respectively, given $\hat{Q}_r^t$ and $\hat{Q}_c^t$, the gradient descent ascent (GDA) update can be written as follows: if $\hat{Q}_l^t(s, a) = \hat{Q}_r^t(s, a) + \lambda_t \hat{Q}_c^t(s, a)$ and $\hat{V}_c^t(\rho) = \sum_{s \in \mathcal{S}} \rho(s) \sum_{a \in \mathcal{A}} \pi_t(a|s) \hat{Q}_c^t(s, a)$, then,

$$\pi_{t+1}(a|s) = \frac{\pi_t(a|s) \exp\left(\eta_1 \hat{Q}_l^t(s, a)\right)}{\sum_{a'} \pi_t(a'|s) \exp\left(\eta_1 \hat{Q}_l^t(s, a')\right)} \quad (6)$$

$$\lambda_{t+1} = \mathbb{P}_{[0, U]}[\lambda_t - \eta_2\, (\hat{V}_c^t(\rho) - b)]. \quad (7)$$

Here $\mathbb{P}_{[a, b]}$ is a projection onto the $[a, b]$ interval, and $\eta_1$ and $\eta_2$ are the step-size parameters for the primal and dual updates respectively. In the tabular setting, the resulting algorithm is the same as that analyzed by Ding et al. (2020).

Analyzing the primal and dual regret for the above updates is fairly standard in online linear optimization. Using results from the paper of Orabona (2019, Theorem 6.8), by setting $\eta_1 = \sqrt{\frac{2\log|\mathcal{A}|}{t}}\frac{1-\gamma}{1+U}$, $\eta_2 = \frac{U(1-\gamma)}{\sqrt{t}}$ and $U = \frac{2}{\zeta(1-\gamma)}$, we get $\mathcal{R}^p(\pi^*, T) \leq \frac{1+U}{1-\gamma}\sqrt{2\log|\mathcal{A}|}\sqrt{T}$ and $\mathcal{R}^d(\lambda, T) \leq \frac{U}{1-\gamma}\sqrt{T}$. Observe that both the primal and dual regret scale as $O(\sqrt{T})$, and using Theorem 3.1, both the average optimality gap and constraint violation will decrease at an $O(1/\sqrt{T})$ rate.

We also note that obtaining the above bounds requires setting the two step-sizes ($\eta_1$ and $\eta_2$) to specific values that depend on problem-dependent parameters. In Figure 1, we have seen that GDA is quite sensitive to the

values of $\eta_1$ and $\eta_2$, even in the simple tabular setting. In order to alleviate this, we use the recent progress in online linear optimization, and propose robust algorithms in the next section.

#### 4.2.2 Coin-betting

Orabona and Pal (2016) and Orabona and Tommasi (2017) propose *coin-betting* algorithms that reduce the online linear optimization problems in Equation (5) to online betting. Unlike adaptive gradient methods like AdaGrad (Duchi et al., 2011) or Adam (Kingma and Ba, 2014) that require setting the initial step-size, *coin-betting algorithms are completely parameter-free.* In this work, we will directly instantiate the regret-minimization algorithms from these works. We first provide some intuition for the coin-betting algorithms.

**Coin-betting algorithms**: Orabona and Pal (2016) shows that the online linear optimization can be viewed as a problem of placing repeated bets (denoted by $x_t$) in round $t$ on the outcomes of unknown adversarial coin flips (denoted by $c_t$). The outcomes of the coin are either heads or tails meaning that $c_t \in \{-1, +1\}$. With our bet $x_t$, we earn an amount $x_t c_t$ in round $t$. Starting with an initial wealth of $\varepsilon_0$, at round $t$ we place bets with a fraction (denoted by $\beta_t$) of the remaining wealth on either heads or tails where $x_t$ becomes $x_t = \beta_t \left( \varepsilon_0 + \sum_{i=1}^{t-1} x_i c_i \right)$. Our goal is to maximize the wealth generated from this process. The coin-betting strategy uses KT estimator (Krichevsky and Trofimov, 1981) which bets $\beta_t = \frac{\sum_{i=1}^{t-1} c_i}{t}$ fraction of the current wealth on the most common outcome observed until time $t$. Orabona and Pal (2016) connects the problem of maximizing wealth to the problem of minimizing the regret in online linear optimization setting. In particular, the authors view the outcome of coin ($c_t$) as the negative of subgradient of the losses (denoted by $g_t$) on current prediction and $x_t$ as the our response at round $t$. Using this reduction, we get a parameter-free 1-d online linear optimization algorithm where we predict

$$x_t = \left( -\frac{\sum_{i=1}^{t-1} g_i}{t} \right) \left( \varepsilon_0 - \sum_{i=1}^{t-1} x_i g_i \right)$$

in round $t$. This problem can be further extended to $d-$dimensional and Learning with Expert Advice (simplex) setting (Orabona and Pal, 2016). For the problem at hand, the iterates $x_t$ correspond to either the policy $\pi_t$ for the primal problem or the Lagrange multipliers $\lambda_t$ for the dual problem.

We now instantiate the algorithm of Orabona and Pal (2016) for updating the policy (primal variables) in the CMDP setting. In order to do this, we define additional variables $w_t$ for each $(s, a)$ pair and iteration $t$. These variables will be computed recursively, and used to compute the policy $\pi_{t+1}$ at iteration $t$. In particular, for $t \geq 1$,

$$w_{t+1}(s, a) = \frac{\sum_{i=0}^t \tilde{A}_l^i(s, a)}{(t+1) + T/2} \left( 1 + \sum_{i=0}^t \tilde{A}_l^i(s, a) \, w_i(s, a) \right)$$

$$\pi_{t+1}(a|s) = \begin{cases} \pi_0(a|s), \text{ if } \sum_a \pi_0(a|s) \, [w_{t+1}(s,a)]_+ = 0 \\ \frac{\pi_0(a|s) \, [w_{t+1}(s,a)]_+}{\sum_{a'} \pi_0(a'|s) \, [w_{t+1}(s,a')]_+}, \text{ otherwise} \end{cases} \tag{8}$$

where, given $\pi_t$, $\tilde{A}_l^t(s, a)$ is equal to

$$\hat{A}_l^t(s, a) \, \mathcal{I}\{w_t(s, a) > 0\} + [\hat{A}_l^t(s, a)]_+ \, \mathcal{I}\{w_t(s, a) \leq 0\}$$

and $\hat{A}_l^t(s, a) = \frac{1-\gamma}{1+U} \left[ \hat{Q}_l^t(s, a) - \left\langle \hat{Q}_l^t(s, \cdot), \pi_t(\cdot|s) \right\rangle \right]$. $\mathcal{I}\{\omega\}$ is the indicator function with value 1 when condition $\omega$ satisfy. For the above calculation, we use the normalized (by $\frac{1+U}{1-\gamma}$) action-value functions that are ensured to lie in the $[0, 1]$ range. The quantity $\hat{A}_l^t(s, a)$ can be interpreted as the (normalized) advantage function for policy $\pi_t$ in the unconstrained MDP with rewards equal to $r(s, a) + \lambda_t c(s, a)$. Observe that the above update does not have any tunable hyperparameters.

Similarly, we use the coin-betting algorithm of Orabona and Tommasi (2017) to update the Lagrange multipliers, instantiating it in the CMDP setting: for $t \geq 1$, if $\sigma(x) := \frac{1}{1+\exp(-x)}$, then,

$$\lambda_{t+1} = \lambda_0 - \beta_t \left[ \frac{1}{1-\gamma} - \sum_{i=0}^t (\lambda_i - \lambda_0) \, (\hat{V}_c^{\pi_i}(\rho) - b) \right],$$

$$\beta_t = (1 - \gamma) \left( 2\sigma \left( \frac{2 \sum_{i=0}^t (\hat{V}_c^{\pi_i}(\rho) - b)}{\frac{1}{1-\gamma} + \sum_{i=0}^t |\hat{V}_c^{\pi_i}(\rho) - b|} - 1 \right) \right) \tag{9}$$

Similar to the primal update, the dual update uses normalized (by $1/1-\gamma$) value functions that lie in the $[-1, 1]$ range, and does not have any tunable parameter. Importantly, these updates result in *no-regret* algorithms meaning that both the primal and dual regret scale as $o(T)$. Specifically, for the primal updates in Equation (8) and the dual updates in Equation (9), the results of Orabona and Tommasi (2017) imply that

$$\mathcal{R}^p(\pi^*, T) \leq \frac{3(1+U)}{1-\gamma} \sqrt{T} \sqrt{1 + \text{KL}(\pi_0||\pi^*)},$$

$$\mathcal{R}^d(\lambda, T) \leq \frac{1}{1-\gamma} + \left\| \lambda - \lambda^0 \right\| \sqrt{\left( \frac{1}{(1-\gamma)^2} + \frac{G_T}{1-\gamma} \right) \Gamma_T},$$

where $\text{KL}(\pi_0||\pi^*) = \mathbb{E}_{s \sim \nu_{\rho, \pi^*}} \text{KL}(\pi_0(\cdot|s)||\pi^*(\cdot|s))$, $\Gamma_T = \log \left( 1 + (G_T(1-\gamma) + 1)^2 \left\| \lambda - \lambda^0 \right\|^2 \right)$ and $G_T = \sum_{i=0}^T |\hat{V}_c^{\pi_i}(\rho) - b| = O(T)$. Since both regrets scale as $O(\sqrt{T})$ in the worst case, using the coin-betting updates will also result in an $O(1/\sqrt{T})$ decrease in both

the average optimality gap and constraint violation. Unlike the updates in Section 4.2.1, the coin-betting updates do not require tuning a hyperparameter.

If we can control the approximation errors, we can use the above algorithms to completely instantiate the primal-dual framework. In Appendix A, we do this for the simpler tabular setting, and consider the linear function approximation setting in the next section.

# 5   PUTTING EVERYTHING TOGETHER

In this section, we will bound the approximation errors in the linear function approximation setting and instantiate the above framework.

In order to scale to large state-action spaces, we consider the special case of linear function approximation and assume *global* access to a $d$-dimensional feature map $\Phi : \mathcal{S} \times \mathcal{A} \to \mathbb{R}^d$. Given $\Phi$, we make the following (approximate) realizability assumption on action-value functions (Abbasi-Yadkori et al., 2019).

**Assumption 5.1** (Linear function approximation)**.** With global access to the feature map $\Phi$, the action-value functions for each memoryless policy $\pi$ are $\varepsilon_{\mathrm{b}}$-close to the span of the state-action features i.e.

$$\inf_{\theta \in \mathbb{R}^d} \max_{(s,a)} |Q_r^\pi(s,a) - \langle \theta, \phi(s,a) \rangle| \le \varepsilon_{\mathrm{b}},$$
$$\inf_{\theta \in \mathbb{R}^d} \max_{(s,a)} |Q_c^\pi(s,a) - \langle \theta, \phi(s,a) \rangle| \le \varepsilon_{\mathrm{b}}.$$

This setting subsumes the tabular case which can be recovered (with $\varepsilon_{\mathrm{b}} = 0$) when $d = |\mathcal{S}||\mathcal{A}|$, and the feature-map consisting of one-hot vectors for each state-action pair. Given a good estimate of $\theta_r^\pi := \arg\min\left[\max_{(s,a)} |Q_r^\pi(s,a) - \langle \theta, \phi(s,a) \rangle|\right]$, we can easily estimate the action-value functions for every $(s,a)$ pair as $Q_r^\pi(s,a) \approx \langle \theta_r^\pi, \phi(s,a) \rangle$. A naive way to estimate $\theta_r^\pi$ is to form a subset $\mathcal{C} \subseteq \mathcal{S} \times \mathcal{A}$ of $(s,a)$ pairs, rollout $m$ independent trajectories using policy $\pi$ and starting from each $(s,a) \in \mathcal{C}$. The average (across trajectories) cumulative discounted return is an unbiased estimate $Q_r(s,a)$ of the action-value function. If $Q_r$ is defined to be the $|\mathcal{C}|$-dimensional vector of estimated action-value functions, and for a fixed set of weights $\omega$ s.t. $\omega(s,a) \ge 0$ and $\sum_{(s,a) \in \mathcal{C}} \omega(s,a) = 1$, we use the weighted-least squares estimate with $z := (s,a)$,

$$\hat{\theta}_r^\pi = \arg\min_\theta \sum_{z \in \mathcal{C}} \omega(z) \left[ \langle \theta, \phi(z) \rangle - Q_r(z) \right]^2. \quad (10)$$

For the $(s,a) \in \mathcal{C}$, the sampling error is $O(1/\sqrt{m})$ by using Hoeffding's inequality. For the $(s,a) \notin \mathcal{C}$, we can then use the resulting $\hat{\theta}_r^\pi$ to estimate $\hat{Q}_r^\pi$ as $\hat{Q}_r^\pi = \langle \hat{\theta}^\pi, \phi(s,a) \rangle$. In Appendix B, we prove the following result to bound the extrapolation errors for all $(s,a)$.

**Lemma 5.2.** *For policy $\pi$, any distribution $\omega$ and subset $\mathcal{C}$, if we use $m$ trajectories to estimate the action-value function for each $(s,a) \in \mathcal{C}$, and solve Equation (10) to compute $\hat{\theta}_r^\pi$, then for any $(s,a) \in (\mathcal{S} \times \mathcal{A})$ pair, the error $|\langle \phi(s,a), \hat{\theta}_r^\pi \rangle - Q_r^\pi|$ can be upper-bounded by*

$$\varepsilon_b (1 + \|\phi(s,a)\|_{G_\omega^\dagger}) + \frac{\|\phi(s,a)\|_{G_\omega^\dagger}}{1 - \gamma} \sqrt{\frac{\log(2|\mathcal{C}|/\delta)}{2m}},$$

*where $G_\omega = \sum_{(s,a) \in \mathcal{C}} \omega(s,a) \phi(s,a) \phi(s,a)^\top$ and $A^\dagger$ is pseudoinverse of $A$.*

Hence, the extrapolation errors can be upper-bounded by choosing $\mathcal{C}$ and $\omega$ to control the $\|\phi(s,a)\|_{G_\omega^\dagger}$ term for each $(s,a)$ pair. Moreover, to ensure scalability, we want that size of $\mathcal{C}$ to be independent of $|\mathcal{S}||\mathcal{A}|$. Fortunately, the Kiefer-Wolfowitz theorem (Kiefer and Wolfowitz, 1960) guarantees the existence of a *coreset* $\mathcal{C}$ s.t. $|\mathcal{C}| \le \frac{d(d+1)}{2}$ and distribution $\omega$ that ensure $\sup_{(s,a)} \|\phi(s,a)\|_{G_\omega^\dagger} \le \sqrt{d}$. If we can find such a $\mathcal{C}$ and distribution $\omega$, then the error, $\tilde{\varepsilon} \le \varepsilon_{\mathrm{b}}(1 + \sqrt{d}) + \frac{\sqrt{d}}{1-\gamma} \sqrt{\frac{\log(2d(d+1)/\delta)}{2m}}$. Here, the first term in error is due to the approximation error ($\varepsilon_{\mathrm{b}}$) and the second term is result of the sampling error (dependent on $m$ trajectories). For our theoretical results, we assume that a coreset $\mathcal{C}$ and distribution $\omega$ is provided, and in Appendix C, we describe the G-experimental design procedure to compute it.

Now that we have control over $\tilde{\varepsilon}$, we instantiate the primal-dual framework with coin-betting algorithms.

## 5.1   CBP ALGORITHM

In this section, we use the coin-betting algorithms ( Section 4.2.2) with linear function approximation to completely specify the Coin-Betting Politex (CBP) algorithm (Algorithm 1). In Algorithm 1, Line 2 computes the coreset $C$ and distribution $\omega$ offline (see Appendix C.2 for details). In order to set $U$, the upper-bound on the dual variables, we need to estimate $\zeta$ and this is achieved by solving the unconstrained problem maximizing $\hat{V}_c^\pi(\rho)$ in Line 3. While this can be done by any algorithm that can solve MDPs with linear function approximation (for example, NPG (Kakade, 2001) or Politex (Abbasi-Yadkori et al., 2019)), we will use Equation (8) (see Section 6) work. After Monte-Carlo sampling $\forall (s,a) \in \mathcal{C}$ (Line 5) and estimating $\hat{\theta}_r^{\pi_t}$ and $\hat{\theta}_c^{\pi_t}$ according to Equation (10) (Line 6), these vectors are used to calculate $\hat{Q}_r^{\pi_t}$ and $\hat{Q}_c^{\pi_t}$ for states encountered in a trajectory generated by policy $\pi_t$ (Line 8). These action-value functions are then used to update the policy at these states. While this can be achieved by any algorithm controlling the primal

regret, CBP uses the parameter-free coin-betting updates (Line 9). At the end of iteration $t$, in Line 11, the dual variables are updated using the coin-betting algorithm.

In the next section, we bound the average optimality gap and constraint violation for CBP.

### 5.1.1  Theoretical Guarantee

We now use Theorem 3.1 to bound the average optimality gap and constraint violation for Algorithm 1. We note that recent work (Liu and Orabona, 2021) uses parameter-free coin-betting algorithms for convex-concave min-max optimization. Since the function to be maximized in Equation (3) is non-concave in $\pi$, this work is not directly applicable to our setting.

**Corollary 5.3.** *Under Assumption 5.1, OG and CV of CBP can be bounded as:*

$$OG \leq \frac{\left( \frac{3(1+U)\sqrt{1+KL(\pi_0||\pi^*)}}{1-\gamma} + \Psi \right)}{(1-\gamma)\sqrt{T}} + \frac{\tilde{\varepsilon}(1+2U)}{1-\gamma},$$

$$CV \leq \frac{\zeta \left( \frac{3(1+U)\sqrt{1+KL(\pi_0||\pi^*)}}{1-\gamma} + \Psi \right)}{\sqrt{T}} + \zeta\,\tilde{\varepsilon}(1+2U),$$

*where $U = \frac{2}{\zeta(1-\gamma)}$, $\tilde{\varepsilon} = \varepsilon_b(1+\sqrt{d}) + \frac{\sqrt{d}}{1-\gamma}\sqrt{\frac{\log(2d(d+1)/\delta)}{2m}}$ and $\Psi = 4U\sqrt{\log((T+1)U)} + 1$.*

Since $U = O(1/1-\gamma)$, the average optimality gap for CBP is $O\left( \frac{1}{(1-\gamma)^3\sqrt{T}} + \frac{\tilde{\varepsilon}}{(1-\gamma)^2} \right)$, while the average constraint violation scales as $O\left( \frac{1}{(1-\gamma)^2\sqrt{T}} + \frac{\tilde{\varepsilon}}{1-\gamma} \right)$. In the function approximation case, ignoring sampling errors, Ding et al. (2020) obtain an $O\left( \frac{1}{(1-\gamma)^3\sqrt{T}} + \left[ \frac{\varepsilon_b}{(1-\gamma)^3} \left\| \frac{d^{\pi^*}}{\rho} \right\|_\infty \right]^{1/2} \right)$ average optimality gap, and an $O\left( \frac{1}{(1-\gamma)^2 T^{1/4}} + \left[ \frac{\varepsilon_b}{(1-\gamma)^3} \left\| \frac{d^{\pi^*}}{\rho} \right\|_\infty \right]^{1/4} \right)$ average constraint violation. Here, $d^{\pi^*}$ is the distribution over states induced by the optimal policy, and $\rho$ is the initial state distribution. Compared to Corollary 5.3, the CV decreases at a slower $O\left(1/T^{1/4}\right)$ rate. Comparing the error terms, the bound for Ding et al. (2020) depends on the potentially large (even infinite) $\left\| \frac{d^{\pi^*}}{\rho} \right\|_\infty$ factor, while forming the coreset ensures that the errors are well controlled for Corollary 5.3. Furthermore, Ding et al. (2020) require knowledge of the typically unknown Slater constant for the CMDP.

On the other hand, Xu et al. (2021) use a neural function approximation (with 1 hidden layer) where only the first layer is trained. In order to compare to Corollary 5.3, we set the width of the second

---

**Algorithm 1:** Coin-Betting Politex

**1** **Input**: $\pi_0$ (arbitary policy initialization), $\lambda_0 \in [0, U]$ (dual variable initialization), $m$ (Number of trajectories), $T$ (Number of iterations), Feature map $\Phi$.

**2** Compute coreset $\mathcal{C}$ and distribution $\omega$

**3** Solve the unconstrained problem $\max_\pi \hat{V}_c^\pi(\rho)$ to estimate $\zeta$ in Lemma 4.1 and set $U = \frac{2}{\zeta(1-\gamma)}$.

**4** **for** $t \leftarrow 0$ **to** $T-1$ **do**

**5** $\quad$ For every $(s,a) \in \mathcal{C}$, use $m$ trajectories starting from $(s,a)$ using policy $\pi_t$ and estimate the action-value functions $q_r(s,a)$ and $q_c(s,a)$.

**6** $\quad$ Compute and store $\hat{\theta}_r^{\pi_t}$ and $\hat{\theta}_c^{\pi_t}$ using Equation (10).

**7** $\quad$ **for** *every $s$ encountered in the trajectory generated by $\pi_t$, and for every $a$* **do**

**8** $\quad\quad$ Compute $\hat{Q}_r^t(s,a) = \langle \hat{\theta}_r^{\pi_t}, \phi(s,a) \rangle$; $\hat{Q}_c^t(s,a) = \langle \hat{\theta}_c^{\pi_t}, \phi(s,a) \rangle$ and $\hat{Q}_l^t(s,a) = \hat{Q}_r^t(s,a) + \lambda_t \hat{Q}_c^t(s,a)$.

**9** $\quad\quad$ Update $\pi_{t+1}(a|s)$ using Equation (8).

**10** $\quad$ **end**

**11** $\quad$ Compute $\hat{V}_c^{\pi_t}(\rho)$, update $\lambda_{t+1}$ using Equation (9).

**12** **end**

---

layer to 1 in Theorem 2 of Xu et al. (2021), making the function approximation equal to a linear mapping with a ReLU non-linearity. In this setting, Xu et al. (2021, Theorem 4) prove that both the average optimality gap and constraint violation scale as $O\left( \frac{1}{(1-\gamma)\sqrt{T}} + \frac{\varepsilon_b}{(1-\gamma)^{2.5}} \left\| \frac{d^{\pi^*}}{\rho} \right\|_\infty \right)$. Observe that although both OG and CV decrease at an $O\left(1/\sqrt{T}\right)$ rate, the error amplification also depends on $\left\| \frac{d^{\pi^*}}{\rho} \right\|_\infty$. Furthermore, this result requires setting the hyperparameters according to the typically unknown $KL(\pi^*||\pi_0)$ quantity. These problems make the theoretical results of Ding et al. (2020) and Xu et al. (2021) potentially vacuous, and the algorithms difficult to use.

## 6  EXPERIMENTS

In this section, we first describe some practical considerations for implementing CBP and compare with baselines GDA and CRPO on a synthetic tabular environment and the Cartpole environment with linear function approximation. For the experiments below, we initialized $\pi_0$ to a random policy and $\lambda_0 = 1$ in Algorithm 1. The parameter $m$ effects the error $\tilde{\varepsilon}$ and the performance. The code can be found at https://github.com/arushijain94/CoinBettingPolitex.

## 6.1 PRACTICAL CONSIDERATIONS

**Checking feasibility and Estimating $\zeta$:** We use the updates in Equation (8) to solve the unconstrained problem maximizing $\hat{V}_c^\pi(\rho)$, and return policy $\tilde{\pi}$. If $\hat{V}_c^{\tilde{\pi}}(\rho) < b$, we declare the problem infeasible, whereas, if $\hat{V}_c^{\tilde{\pi}}(\rho) > b$, we estimate $\zeta = \hat{V}_c^{\tilde{\pi}}(\rho) - b$.[2] It is important to note that Lemma 4.1 does not require the exact maximization of $\hat{V}_c^\pi(\rho)$ to upper-bound $\lambda^*$. Any feasible policy for which $V_c^\pi(\rho) > b$ can be used to estimate $\zeta$ and upper-bound $\lambda^*$, though the tightest upper-bound is obtained for $\max_\pi V_c^\pi(\rho)$ (see the proof of Lemma 4.1 in Appendix B).

**Gradient normalization and practical coin-betting:** Recall that the coin-betting algorithms in Section 4.2.2 require normalizing the gradients by $1+U/1-\gamma$. Unfortunately, this upper-bound on the gradient norms is quite loose in practice, and directly using the updates Equations (8) and (9) results in poor empirical performance. Since coin-betting algorithms do not have a step-size that can be scaled to counteract the normalization, this issue needs to be handled differently. In particular, we continue to directly use the updates in Equation (8) with the normalization, but use a heuristic, Algorithm 2 of Orabona and Tommasi 2017, for updating the dual variable. This heuristic is a way to adaptively normalize the dual gradients (depending on the previously observed values). For the details, see Algorithm 1 in Appendix C.1. While this heuristic introduces a hyperparameter in the dual updates, our empirical results suggest that the resulting coin-betting algorithm is quite robust to the choice of this parameter and so we use this method in our subsequent experiments.

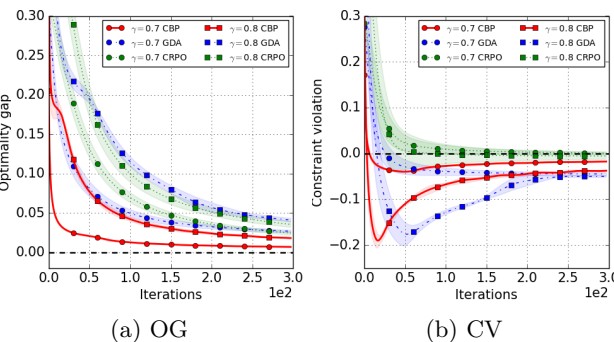

(a) OG       (b) CV

Figure 2: **Environment Misspecification in model-based tabular setting with varying $\gamma$:** Assuming access to the true CMDP, we vary discount factor $\gamma = \{0.7, 0.8\}$. We use the hyperparameters for the original CMDP with $\gamma = 0.9$. CBP converges faster with a smaller variance as compared to GDA and CRPO.

---

[2]If $\hat{V}_c^{\tilde{\pi}}(\rho) = b$, then we return policy $\tilde{\pi}$ as the optimal feasible policy in the CMDP.

## 6.2 TABULAR SETTING

We consider a synthetic gridworld environment similar to Sutton and Barto (2018, Example 3.5) (see Appendix C.3 for details) and set the discount factor $\gamma = 0.9$. We first consider a **model-based setting**, where we have complete knowledge of the CMDP. In Figure 1 (in Section 1), we compared the performance of the three algorithms. For each algorithm, the hyperparameter range is described in Appendix D and the *best hyperparameter* corresponds to the least `OG` while satisfying `CV` $\in [-0.25, 0]$. The key observation is that *CBP is robust to its hyperparameter values*, while GDA and CRPO are sensitive to their hyperparameter values. In Figure 7 (Appendix D.1), we show best performing variants for all methods. In addition, we demonstrate the poor performance of GDA when used with the theoretical step-sizes suggested in Corollary A.1. Next, we measure the robustness of the algorithms with respect to *environment misspecification* where we vary $\gamma$. In Figure 2, we observe that CBP has consistently faster convergence with a lower variation in the performance.

In the **model-free setting**, from Figure 3 we observe the effect of increasing the number of samples in approximating the $Q$ value function on the performance. CBP consistently converges faster than its counterparts in sampling based approaches. In Appendix D.1, we demonstrate CBP robustness to hyperparameters (Figure 8) and environment misspecification (Figure 9).

## 6.3 LINEAR SETTING

In the following experiments, all the algorithms require $O(d)$ memory to construct $Q$ value functions and have a similar handle on the policy $\pi$.

**Gridworld environment:** We start with linear function approximation (LFA) on the gridworld environment. We use tile coding (Sutton and Barto, 2018) to construct $d$-dimensional feature space (see Appendix D.2 for details). We used LSTDQ (Lagoudakis and Parr, 2003) to estimate $Q$ functions with 300 samples for all $(s, a)$ pairs. In Figure 4, we show the performance of the best hyperparameter (see Table 2 for specific values) for each algorithm. We observe that the `OG` of CBP converges consistently faster across different feature dimensions. Again, we observe a good hyperparameter robustness of CBP in Figure 10 (Appendix D). Figure 11 in Appendix D.2 shows that we can obtain similar performance by using G-experimental design, but at a much lower computational cost.

**Cartpole environment with exploration:** We use the Cartpole environment from the OpenAI

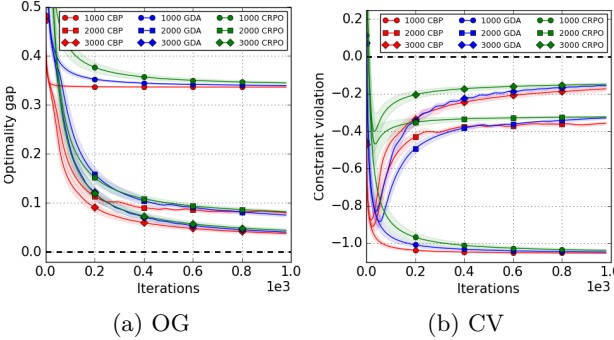

(a) OG                (b) CV

Figure 3: **Effect of sampling in model-free tabular setting:** Number of samples $= \{1000, 2000, 3000\}$ are varied for $Q$ value estimation to observe the change in performance (averaged over 5 runs). The performance improves with increase in samples and CBP converges faster than the baselines GDA and CRPO.

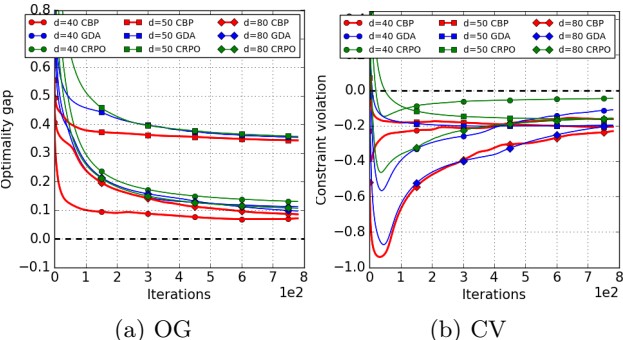

(a) OG                (b) CV

Figure 4: **LFA in gridworld environment:** For varying feature dimension $d$, OG for CBP consistently converges faster than the baselines GDA and CRPO.

gym (Brockman et al., 2016), and modify it to include multiple constraints. The agent is rewarded to keep the pole upright, whereas it receives a constraint reward if (1) the cart enters certain areas (x-axis position), or (2) the angle of pole is smaller than a certain threshold (see Appendix D.2 for details). We used tile coding to construct the feature space, and LSTDQ to estimate the $Q$ functions for both reward and constraint reward.

In Figure 5 we show the cumulative discounted reward and the constraint violation (CV 1, CV 2) for the two constraints as mentioned above. The dark lines correspond to the best hyperparameter that achieves the maximum return, while satisfying CV $\in [-6, 0]$ for both constraints, with the lighter shade-lines correspond to the other hyperparameters. All the algorithms satisfy the constraints and achieve comparable reward, but CBP has considerably less variance in performance for different values of the hyperparameters. In Figure 12 (Appendix D.2), we added entropy regularization (Geist et al., 2019; Haarnoja et al., 2018) and observed a similar robustness for CBP.

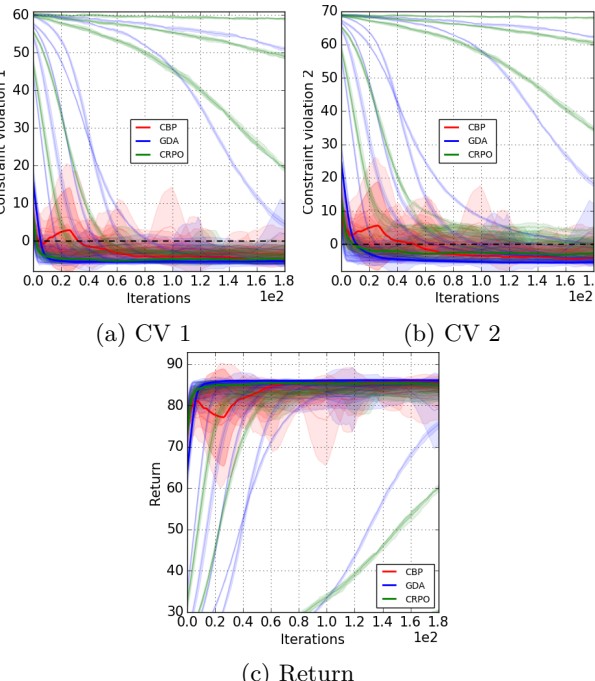

(a) CV 1                (b) CV 2

(c) Return

Figure 5: **Cartpole environment:** Performance of CBP, GDA and CRPO with two constraints (averaged across 5 runs). The dark lines depict performance with the best hyperparameters. Light lines correspond to performance with other setting of hyperparameters. CBP exhibit robustness to the choice of hyperparameters.

# 7  CONCLUSION

In this paper, we proposed a general primal-dual framework to solve CMDPs with tabular and linear function approximation setting. The main motivation of this work was to reduce the hyperparameter sensitivity in the policy optimization setting. We empirically showed that the existing algorithms suffer from high hyperparameter sensitivity (Figure 1). Furthermore, they can even lead to uncontrolled errors in function approximation setting. To alleviate the above mentioned problems, we proposed a theoretically sound CBP algorithm which leverages the coin-betting technique from online linear optimization. In addition, we also use experimental design procedure to control the errors.

Orabona and Tommasi (2017) has shown coin-betting algorithms scale to neural networks. Similarly, in future we plan to scale CBP to non-linear function approximation. We aim to use the recent advances in online linear optimization to design "painless" parameter-free policy optimization algorithms. We believe that this is important for reproducibility in RL and hope our work will encourage future research in this area.

## Acknowledgements

We would like to thank Tor Lattimore for feedback on the paper. Csaba Szepesvári acknowledges the funding from Natural Sciences and Engineering Research Council (NSERC) of Canada and "Design.R AI-assisted CPS Design" (DARPA) project. Doina Precup and Csaba Szepesvári both acknowledge funding from Canada CIFAR AI Chairs Program for Mila and Amii respectively.

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
