# OpenReview forum: "Towards Painless Policy Optimization for Constrained MDPs"
_auai.org/UAI/2022/Conference — UAI 2022 Poster_

### Official Review · Reviewer_zTC8 · 2022-04-02

**Q2(1) Originality/Novelty:** 3
**Q2(2) Significance/Impact:** 3
**Q2(3) Correctness/Technical Quality:** 4
**Q2(6) Clarity Of Writing:** 3
**Q6 Overall Score:** 8
**Q8 Confidence In Your Score:** 4

**Q1 Summary And Contributions:**

In this article, the authors propose an original RL algorithm for approximately solving Constrained MDPs.
They put together some recent techniques and add several new theoretical results.
The CBP algorithm is described and extensive experiments are presented.
Unfortunately, the code of the algorithm/experiments are not made available.

**Q2 Assessment Of The Paper:**

More detailed information regarding each of these aspects is given below:

**Q2(4) Quality Of Experiments (Optional):**

3: Good: The experimental evaluation is adequate, and the results convincingly support the main claims.

**Q2(5) Reproducibility:**

3: Good: Key resources (e.g., proofs, code, data) are available and key details (e.g., proofs, experimental setup) are sufficiently well-described for competent researchers to confidently reproduce the main results.

**Q3 Main Strengths:**

Numerous "strong" theoretical results.
Still an "intuitive" RL algorithm for CMDP.

**Q4 Main Weakness:**

Unavailability of code and non-reproducibility of experiments.

**Q5 Detailed Comments To The Authors:**

This paper provides a quite involved mathematical analysis of an original primal-dual type RL algorithm for CMDPs.
In addition to its mathematical strength, it provides good algorithmic description and strong experiments.
The only weak point of this paper is the lack of reproducibility of the experimental results. Even though it is mainly a theoretical contribution, the reproducibility of experimental results seems to me crucial, in the domain of RL.
This written, I find that the paper is still very good.


**Q7 Justification For Your Score:**

The contribution is very strong and the algorithm useful. for me, this is really a good paper.
My only reservation is about the unavailability of the code: My feeling is that code availability is nowadays as important as proofs availability, to assess algorithmic advances. I hope the authors will made their code available...

**Q9 Complying With Reviewing Instructions:**

1: Yes.

---

### Official Review · Reviewer_daGz · 2022-04-09

**Q2(1) Originality/Novelty:** 2
**Q2(2) Significance/Impact:** 2
**Q2(3) Correctness/Technical Quality:** 3
**Q2(6) Clarity Of Writing:** 2
**Q6 Overall Score:** 5
**Q8 Confidence In Your Score:** 2

**Q1 Summary And Contributions:**

The paper describes a primal-dual framework for solving constrained MDPs by online linear function optimization based on an online betting algorithm (Coin Betting Politex or CBP) that doesn’t have hyper parameters that need to be tuned. (By contrast, hyper parameters are commonly used by gradient descent-ascent methods.) The CBP algorithm is proved sound and bounds are given.

**Q2 Assessment Of The Paper:**

More detailed information regarding each of these aspects is given below:

**Q2(4) Quality Of Experiments (Optional):**

2: Fair: The experimental evaluation is weak: important baselines are missing, or the results do not adequately support the main claims.

**Q2(5) Reproducibility:**

3: Good: Key resources (e.g., proofs, code, data) are available and key details (e.g., proofs, experimental setup) are sufficiently well-described for competent researchers to confidently reproduce the main results.

**Q3 Main Strengths:**

The approach is mathematically sophisticated and appears to be very carefully thought through.


**Q4 Main Weakness:**

The paper is written in a way that makes it difficult to understand or appreciate the contribution, and put it in perspective. In addition, the test problems are very small with no discussion of scalability.

**Q5 Detailed Comments To The Authors:**

The name “painless policy optimization” is used in the title, without explanation in the paper. My guess is that it means the approach minimizes hyper-parameter tuning. If so, that should be stated clearly. (My first guess was that the adjective “painless” is used ironically!)

Although difficult to read, the paper is in a sense also well-organized and carefully and precisely written. What I think is missing is that it does not place the results in perspective and provide a high-level explanation that can help the reader better understand the significance of the results. Is it all about reducing use of hyper-parameters in online optimization? I'm guessing that's it, but am not sure, or even sure that's important compared to simplicity or scalability (for example) -- I would have liked a clearer explanation.

The paper overly relies on long supplementary appendices, not just for proofs, but for details that should be in the paper itself. I wanted to ignore the appendices, but couldn’t, because I needed to see the details of the example problems to help interpret the results. Repeated references to supplementary appendices may be unavoidable, I don’t know, but it certainly makes it more difficult for the reader.

The test problems are surprisingly small: a 5 by 5 grid world and a cart pole problem. When test problems are this small, I think it’s important to include some discussion of scalability. Is it really true that the research community cannot solve completely observed constrained MDPs bigger than a 5 by 5 grid world?

The graphs are almost too small to read.

I missed some discussion of future work, potential applications, or any attempt to place this work into a wider perspective that could help the reader better understand its significance.

p. 4 typo: “our result does it require”

**Q7 Justification For Your Score:**

A mathematically sophisticated approach to online linear optimization for constrained MDPs that seems technically correct (although it is beyond my ability to verify the details). The paper is difficult to read, however, and the authors do not do enough to help the reader place this work in perspective and understand its significance.

**Q9 Complying With Reviewing Instructions:**

1: Yes.

---

### Official Review · Reviewer_thBS · 2022-04-11

**Q2(1) Originality/Novelty:** 3
**Q2(2) Significance/Impact:** 3
**Q2(3) Correctness/Technical Quality:** 3
**Q2(6) Clarity Of Writing:** 3
**Q6 Overall Score:** 4
**Q8 Confidence In Your Score:** 3

**Q1 Summary And Contributions:**

This paper focused on the discounted constrained Markov decision process (CMDP) and proposed the Coin Betting Politex (CBP) with $O(1/(1-\gamma)^3\sqrt{T})$ reward-suboptimality guarantee and $O(1/(1-\gamma)^2\sqrt{T})$ constraint violation. In addition, the experiment results also support the efficiency of this novel algorithm.

**Q2 Assessment Of The Paper:**

More detailed information regarding each of these aspects is given below:

**Q2(4) Quality Of Experiments (Optional):**

3: Good: The experimental evaluation is adequate, and the results convincingly support the main claims.

**Q2(5) Reproducibility:**

3: Good: Key resources (e.g., proofs, code, data) are available and key details (e.g., proofs, experimental setup) are sufficiently well-described for competent researchers to confidently reproduce the main results.

**Q3 Main Strengths:**

1. Compared with prior works, the Coin Betting Politex does not require tuning a hyperparameter, which is much more stable than other algorithms.
2. This work provides a theoretical guarantee for algorithm performance. In addition, Experiment results also show that the novel algorithm outperforms other baseline algorithms.

**Q4 Main Weakness:**

1. In the abstract, the author mentions that it is possible to find a policy with $O(1/(1-\gamma)^3\sqrt{T})$ reward-suboptimality guarantee and $O(1/(1-\gamma)^2\sqrt{T})$ constraint violation. However, there lacks the main theorem to illustrate this result. It is unclear how to set the parameter $\pi_0,\lambda_0,m$ in the Coin-Betting Politex. It is difficult to check the claim that the Coin Betting Politex algorithm does not require tuning this parameter.
2. For all works on linear function approximation, the dimension $d$ is important and relevant to the final result. However, if we assume the approximation error $\epison_b=0$, the result in the abstract shows that the output algorithm has the same guarantee whatever the dimension $d$ is, which conflicts with priors work on linear function approximation.
3. This work lacks an estimation for the complexity, and it is better if there exists a comparison with the prior results on both time and space complexity.

**Q5 Detailed Comments To The Authors:**

See Q4

**Q7 Justification For Your Score:**

There is no main theorem to describe the main result, and it is not clear how to set the parameters. Without this information, it is difficult to check the main contribution that the Coin Betting Politex algorithm does not require tuning this parameter.


**Q9 Complying With Reviewing Instructions:**

1: Yes.

---

### Decision · Program_Chairs · 2022-05-15

**Decision:**

Accept (Poster)

**Comment:**

Meta Review: The theoretical merit are recognized by reviewers. There are also several drawbacks of the paper, as pointed out by reviewers (and not well-addressed in the review response), which are
* How to set the hyperparameters in the Coin Betting Politex algorithm
* The impact of the dimension d
* The content is too dense, so the paper is hard to follow.